# Quasi-equilibrium phase coexistence in single component supercritical fluids

Seungtaek Lee [1], Juho Lee[1], Yeonguk Kim[1], Seokyong Jeong[1], Dong Eon Kim[1,2]✉ & Gunsu Yun [1,2,3]✉

In their supercritical state simple fluids are generally thought to assume a homogeneous phase throughout all combinations of pressures and temperatures, although various response functions or transport properties may exhibit anomalous behavior, characterizing a state point as either more gas-like or liquid-like, respectively. While a large body of results has been compiled in the last two decades regarding the details of the supercritical phase in thermodynamic equilibrium, far less studies have been dedicated to out-of-equilibrium situations that nevertheless occur along with the handling of substances such as carbon dioxide or Argon. Here we consider successive compression-expansion cycles of equal amounts of Argon injected into a high-pressure chamber, traversing the critical pressure at two times the critical temperature. Due to expansion cooling, the fluid temporarily becomes sub-critical, and light scattering experiments show the formation of sub-micron-sized droplets and nanometer-scale clusters, both of which are distinct from spontaneous density fluctuations of the supercritical background and persist for a surprisingly long time. A kinetic rate model of the exchange of liquid droplets with the smaller clusters can explain this behavior. Our results indicate non-equilibrium aspects of supercritical fluids that may prove important for their processing in industrial applications.

[1] Department of Physics, Pohang University of Science and Technology, Pohang, Republic of Korea. [2] Max Planck Center for Attosecond Science, Max Planck POSTECH/KOREA Research Initiative, Pohang, Republic of Korea. [3] Division of Advanced Nuclear Engineering, Pohang University of Science and Technology, Pohang, Republic of Korea. ✉email: kimd@postech.ac.kr; gunsu@postech.ac.kr

Supercritical fluids (SCFs), first discovered in 1822 by Charles Cagniard de la Tour[1], have attracted continuous scientific and industrial interests owing to their unique properties such as higher density compared to the gas phase, lower viscosity compared to the liquid phase, and intrinsically high level of density fluctuations[2,3]. The lower viscosity, higher diffusivity, and superior solubility compared to liquid counterparts mean that SCFs can enable enhanced chemical reaction rates[4]. Some of the applications of SCFs that have been extensively investigated include the selective extraction of molecules from natural materials[5] and the synthesis of polymer nanocomposite foams[6,7]. SCFs are used in heat exchange systems[8] and as reaction media for chemical processes[9].

It is a common belief that there is only one phase in SCFs, unlike the subcritical fluids that have distinct liquid and vapor phases. SCFs do not have any surface tension or phase transition, because, far from the critical point, they are homogeneous and structureless. However, over the last two decades, it was found that SCFs may exhibit more liquid-like (LL) or gas-like properties as pressure, temperature, or density are varied across appropriate regions in state space[10–18]. It must be emphasized that these qualifications always refer to specific characteristics such as sound dispersion[10,11], unexpected variations in thermodynamic response functions[12], or correlations in density fluctuations[19], and always imply continuous variations of all relevant observables[20].

However, many industrial processes involve strong nonequilibrium dynamics of SCFs, where the spatial and temporal rates of change are significant. The examples include meteorological flows in the atmosphere of Venus, SCF extraction[21], and high-pressure flows in propulsion systems. In the latter, the fluid fuel stream injected into an ambient fluid undergoes a combination of continuous changes (in thermodynamic and transport properties) and discrete changes (formation of ligaments and droplets), depending on the ambient pressure (chap. 10 in[22]). In the intermediate (transcritical) pressure regime, the fuel stream may form a fractal-like boundary where the notion of two-phase and one-phase becomes blurry[23].

This suggests that the investigation of the transient behavior of a SCF under strong nonequilibrium conditions, subject to sudden disturbances may call for a more systematic study of its own. Here we carry out controlled compression–expansion cycles of argon as an archetypal inert species. We clearly observe through light scattering experiments that after initial rapid expansion and cooldown liquid submicron-sized droplets and nanometer scale clusters are formed under subcritical conditions. These float in the chamber and persist as metastable LL compact fluid packages for a surprisingly long time until they are absorbed into the gaslike supercritical background[23]. Across the critical pressure, the lifetime of the LL droplets first decreases and then increases. The opacity of the fluid volume, however, is found to be almost exclusively due to the small clusters. Because of the rather long lifetime of clusters and droplets, we suggest a kinetic model assuming quasi-steady state conditions that takes the mass exchange of clusters and droplets into account and can explain the observed phenomenology.

## Results

**Experimental apparatus and conditions**. A reciprocating compressor delivers argon fluid into the high-pressure chamber through a check valve (opening pressure: 300 bar). In each cycle, the compressor supplies about 6.8 cm$^3$ of argon fluid (under the standard condition). A series of experiments has been carried out along the line of red points shown in Fig. 1d (the extended phase diagram of supercritical argon)[24,25]. Because the temperature is

about two times the critical temperature, spontaneous critical phenomena such as density fluctuations do not occur; however, the fluid undergoes a sudden expansion and adiabatically cools to form a liquid during the compression. When the argon fluid reaches below 140 K, it undergoes the phase change and the liquid droplets and clusters are generated. The temperature decrease of an expanding fluid is well understood[26–28], and we provide our experimental results compared with the COMSOL Multiphysics simulations in the following section.

The imaging optics with the ICCD camera captures images and videos of droplets with two different magnifications (×2.5 and ×11). The droplets are individually identified owing to their higher Rayleigh–Mie scattering intensity compared to the scattering intensity of the argon fluid and clusters in the background. The number density and the mean size of droplets are obtained from the lower- and higher-magnification data, respectively. Note that the size of individual droplets cannot be measured by direct imaging because of the diffraction limit of the visible light (light source: 633 nm He–Ne laser). Hence, Brownian motion analysis is adopted to determine the mean size of the droplets.

**Temperature drop of expanding fluids**. To demonstrate that the expanding argon fluid reaches a low enough temperature to become a liquid, we perform fluid/heat transport simulations (using the multiphysics simulation tool COMSOL) and compare with the experiments. As the compressor delivers a relatively small amount of fluid at each cycle, a thermocouple of finite heat capacitance may not experience a measurable cooling. Thus, we set up a temperature measurement scheme with a small buffer chamber filled up with 300 bar of argon fluid and suddenly purge out to measure the temperature drop by a thermocouple (Fig. 2a, b). Figure 2b shows the temperature profile along the axis. In the experiment, the thermocouple is placed at 20–60 mm away from the valve exit. Each simulation result is averaged for 0–200 ms within the red dashed area to compare with the experiment.

Another set of simulations shows the temperature profile when the fluid undergoes a sudden expansion through the compressor's check valve along the pipe that connects to the main chamber (Fig. 2c, d). The gradient in the temperature profile decreases as the initial pipe pressure increases. Figure 2d shows the minimum temperature reached within the simulation volume by varying the pipe ($z > 0$) pressure. The fluid reaches low enough temperature to form a liquid when the pipe pressure is below 100 bar, which agrees with the trend of generation of droplets and clusters in Fig. 3a, c.

Although the local fluid temperature reaches very low, the overall temperature inside the chamber stays unchanged from the room temperature owing to the conductive heat transport at the pipe surface. Using the simplified dimensions of the stainless steel pipe ($r \sim 2$mm, $l \sim 800$mm, and $d \sim 10$mm, where $r$, $l$, and $d$ are the inner diameter, the total length, and the thickness of the pipe, respectively) and the thermal conductivity of a stainless steel $k \sim 16$W · m$^{-1}$K$^{-1}$ and assuming the temperature difference between the inside and outside of the pipe $\delta T \sim 10$K, the heat transfer rate becomes

$$\frac{dQ}{dt} \sim k\left(\frac{2\pi r l}{d}\right)\delta T \sim 160\text{W} \tag{1}$$

In the experiment, the compressor delivers $N_1 \sim 2.7 \times 10^{-4}$mol · cycle$^{-1}$, and operates in $f \sim 4.1$Hz. Thus, the number of particles compressed into the chamber per unit time is $N = N_1 \cdot f \sim 1.1 \times 10^{-3}$mol · s$^{-1}$. Considering the power

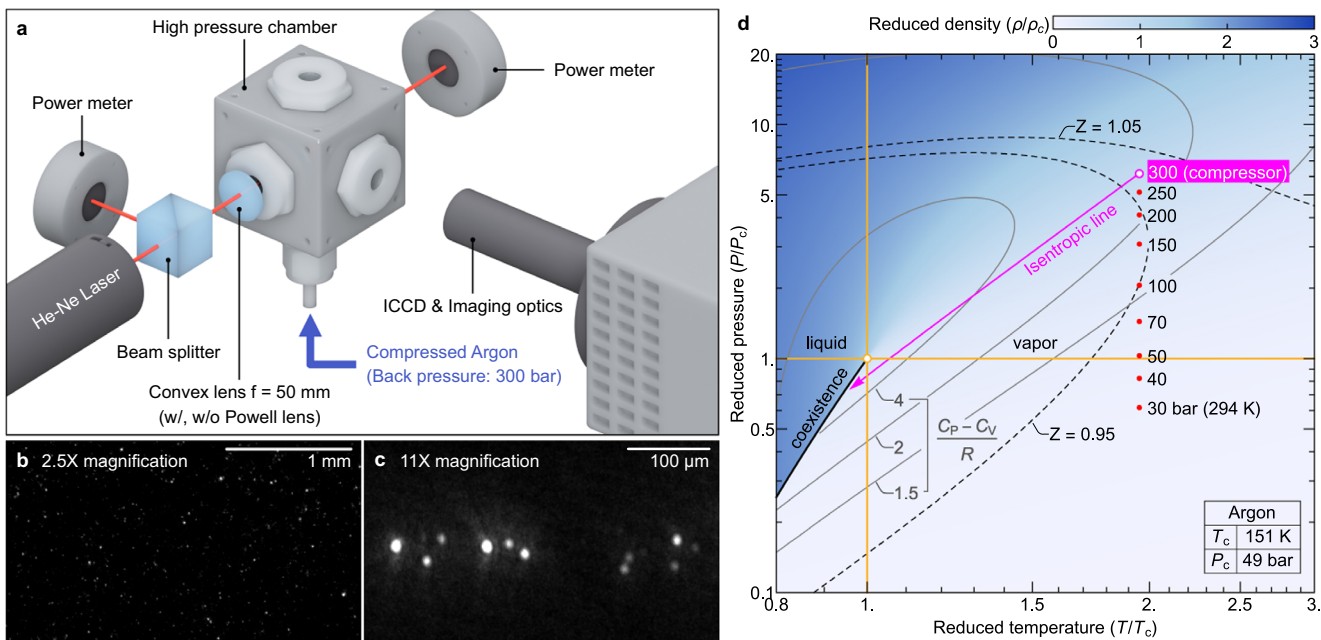

**Fig. 1 Experimental apparatus and conditions. a** Schematic of the high-pressure chamber system. The incident He–Ne laser (633 nm, 5 mW) has a beam waist radius of about 50 μm at the focus. The incident laser is additively reshaped with a laser line generator (called the Powell lens) for the lower magnified imaging optics. Imaging optics and intensified CCD (ICCD) capture the Rayleigh–Mie scattering signals from individual droplets. **b, c** Images of droplets with ×2.5 and ×11 magnifications at 100 bar. **d** Extended phase diagram of homogeneous and equilibrium argon fluid. The red points indicate the experimental conditions. The working pressure of the compressor is 300 bar. The argon fluid undergoes an adiabatic expansion to form the liquid droplets and clusters. The compressibility factor $Z = P/\rho RT$ which is unity for the case of an ideal gas. The thermophysical properties of argon are obtained from the NIST Chemistry WebBook[24].

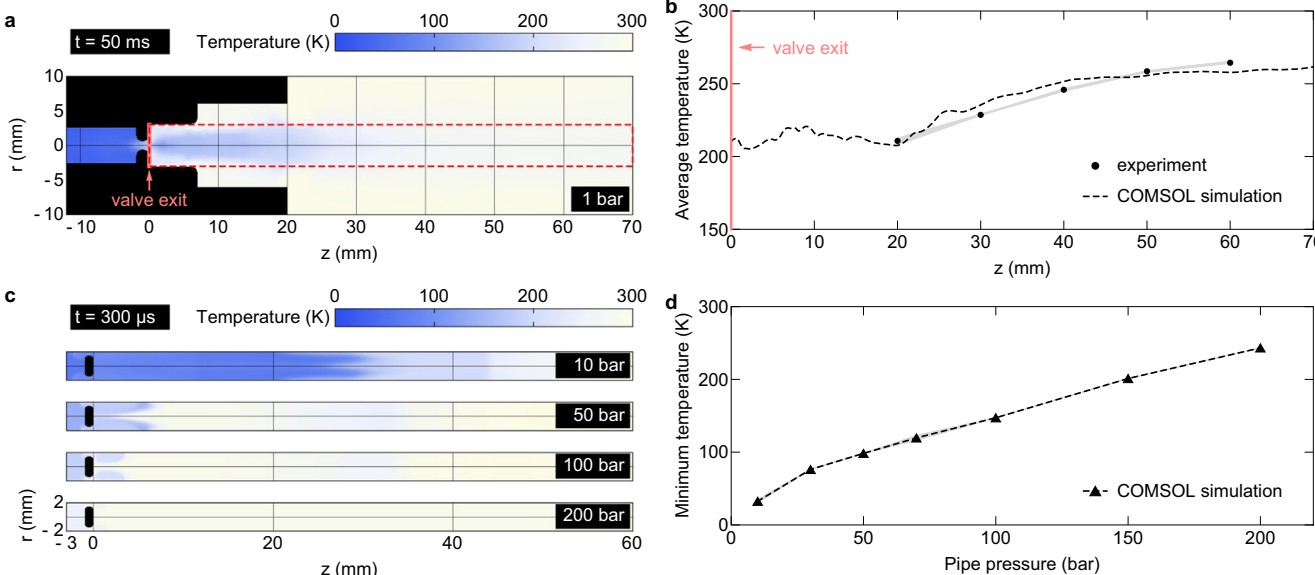

**Fig. 2 A temperature drop of an expanding fluid by COMSOL Multiphysics simulation compared with the experiments. a** The buffer chamber with 300 bar suddenly expands to an open space having 1 bar. **b** The experimental measurement is compared with the time- and space-averaged temperature in the simulation results (0–200 ms, red dashed area). **c** A set of simulations shows the fluid expansion through the check valve of the compressor outlet to the pipe connecting to the main chamber as a function of the initial pipe pressure. **d** Minimum temperatures for different pipe pressures. The error bands are the standard deviations of each point.

required to raise the temperature of such fluid from 100 to 300 K,

$$P_{\text{heat}} = \bar{C}_P N \Delta T \sim 5.5\text{W} \tag{2}$$

where the average heat capacitance $\bar{C}_P \sim 25\text{J} \cdot \text{K}^{-1}\text{mol}^{-1}$ within the given temperature range under 10 bar[24]. The conductive heat transfer rate through the pipe is much greater than the heating power, which keeps the chamber at room temperature. This result is also confirmed in the COMSOL simulation. Further compression to higher pressure increases convective heat transport, which helps keep the temperature of the fluid in the chamber.

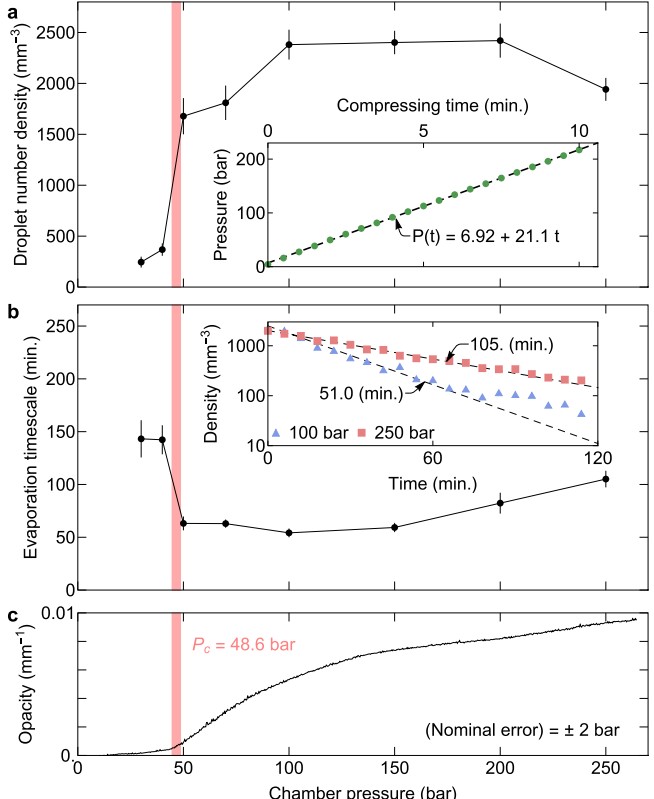

**Fig. 3 Presence of droplets and clusters. a** The number density of droplets generated by the continual compression of argon fluid in the high-pressure chamber. The droplet number density does not increase as the pressure exceeds 100 bar because the temperature drop is not large enough to form a liquid at a higher chamber pressure (i.e., a smaller pressure difference between the chamber and check valve) and the droplets evaporate. The number density of droplets decreases due to evaporation in the medium. **b** The evaporation timescale at different pressures. **c** The major contribution to medium opacity comes from clusters. The error bars are the standard deviations of each point.

**Formation of droplets and clusters**. Figure 3 shows how the SCF properties change with the increase of pressure in terms of droplet density, the average lifetime of droplets, and medium opacity. We note the dramatic changes of these properties across the critical pressure. Figure 3a shows the sudden increase of the droplet density across the critical pressure, above which it is saturated. The argon gas is cumulatively compressed into the chamber through the check valve while creating liquid droplets and clusters. The droplet generation rate decreases as the chamber pressure increases, and eventually, the droplet formation stops. It is consistent with the COMSOL simulation results—a sufficient temperature drop to form a liquid during a sudden expansion occurs when the chamber pressure is below 100 bar (Fig. 2d). The droplets produced during the compression also undergo evaporation within the fluid. The number density of droplets does not keep increasing throughout the compression, as shown in Fig. 3a. Instead, the generation and evaporation of the droplets occur simultaneously and compete, and finally, above 100 bar, evaporation dominates.

The lifetime of droplets or the evaporation timescale (measured by the time when the droplet number density becomes 1/10th of the maximum) at different chamber pressures is shown in Fig. 3b. For each pressure condition, the droplet number density is tracked for 2 h after the compression. As the chamber pressure increases above the critical pressure, the evaporation timescale

suddenly decreases and then increases. In the following section, we explain both the discontinuous change near the critical pressure and the monotonic elevation at higher pressures based on the evaporation model and the resulting cluster effect.

As shown in Fig. 3c, the opacity inside the chamber rapidly increases, which we attribute to the generation of a large number of clusters. Note that these clusters are not spontaneously produced by critical phenomena but by a compression–expansion process.

Unlike droplets, the clusters are too small to be individually identified. They increase the medium opacity through the Rayleigh scattering and absorption. To clarify that the major contribution to the opacity is from these clusters, rather than the droplets, we estimate the opacity due to the scattering by the particles:

$$\chi(\lambda) = \sum_s n_s \sigma_s(\lambda) \qquad (3)$$

where $n_s$ and $\sigma_s$ are the number density and the total scattering cross-section of the particle species $s$ (droplets, clusters, and atoms), respectively[29–31], and $\lambda$ is the wavelength of the incident light. For a large particle with the size comparable to $\lambda$, $\sigma \sim \pi r^2$ where $r$ is the mean radius of the particles. The upper limit of the opacity caused by the droplets is about 0.002 mm$^{-1}$ (using $n_d \sim$ 2500 mm$^{-3}$ around 150 bar and $r_d \sim 0.5 \times 10^{-3}$ mm, see Fig. 3), which is quite small, implying that the major contribution to opacity comes from the clusters and not from the droplets.

The Rayleigh scattering cross-section of small particle with radius $r$ is

$$\sigma = \frac{2\pi^5}{3} \frac{(2r)^6}{\lambda^4} \left(\frac{n^2 - 1}{n^2 + 2}\right)^2 \qquad (4)$$

where $\lambda$ is the wavelength of the incident light and $n$ is the refractive index ratio between the particle and a medium[31]. Figure 4 shows the conditions of the cluster size and number density to achieve such opacity. The gray area is where the mass conversation is violated for the given condition (100 bar, 294 K). A cluster is assumed to have $n \sim 1.2$—the value for a liquid argon at $\sim 100$K[32]. The cluster conditions should lie in the yellow area to explain the measured opacity. For the case of 'atoms only', the chamber opacity is not even close to the experimental result (see the green point in Fig. 4). Thus, the clusters are the leading cause of the fogginess and opacity of the medium.

It turns out that a large number of clusters in the argon fluid also has a significant effect on the mass transport at the droplet surface, as we discuss below. The clusters in the SCF affect the evaporation timescale of the droplets.

**Droplet size**. A large number of submicron-size droplets floating in the argon SCFs are visually identified. Notably, the the droplets persist for a long time, which enabled us to apply the direct imaging and the Brownian motion analysis to measure the size.

The mean size of droplets is measured by considering them as Brownian particles floating in the argon fluid (see Supplementary Movie 1). The small displacements of droplets from their jiggling motion are tracked by high-resolution imaging optics (×11 magnification) and ICCD (50 frames per second). The videos are captured 3 min after the compression allowing any turbulence in the medium to be stabilized at each pressure condition.

The mean squared displacement $\sigma^2$ of Brownian particles for one dimensional motion is expressed by the following equation[33]:

$$\sigma^2 = 2Dt \qquad (5)$$

where $D$ is the diffusion coefficient of the surrounding medium and $t$ is the elapsed time (in our experiment, $t = 0.02$s). The

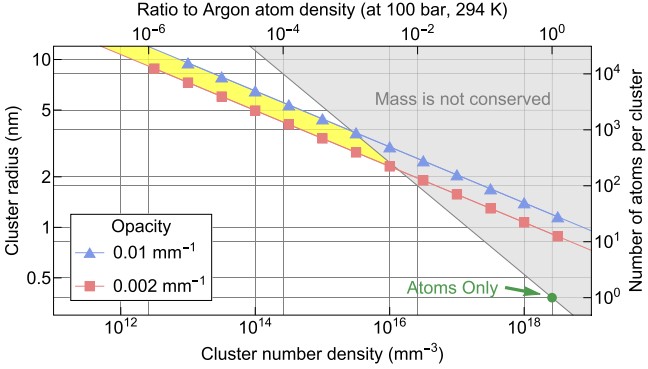

**Fig. 4 Expected number density and size of the clusters to explain the opacity.** The gray shaded area is forbidden because the number of atoms exceeds the total number of atoms for the given chamber condition (100 bar, 294 K). The clusters should have density and size within the yellow shaded area. The green dot corresponds to the opacity for the case of atoms only (i.e., no clusters).

diffusion coefficient is related to the mean radius $r$ of the Brownian particles through the Stokes–Einstein equation[33]:

$$D = \frac{k_\mathrm{B} T}{6\pi\eta r} \qquad (6)$$

where $k_\mathrm{B}$, $T$, and $\eta$ are the Boltzmann constant, the temperature, and the dynamic viscosity of the surrounding medium, respectively. We referred to the NIST Chemistry WebBook[24] for the dynamic viscosity of argon at each pressure. The mean radius of the droplets, based on the dynamic viscosity, varies from about 250–400 nm depending on the final chamber pressure, as shown in Fig. 5. However, if the medium contains a large number of clusters, using a homogeneous medium viscosity may introduce errors in size estimation. When some portions of the medium are clustered, the viscosity tends to be lower than that of the homogeneous medium, and as a result, it leads to an underestimation of the droplet size. We will discuss this further in the discussion section.

**Droplet lifetime: cluster effect on evaporation.** The trend of the estimated droplet radii around the critical pressure (Fig. 5) appears to contradict the trend of the droplet lifetime (the evaporation timescale) (Fig. 3b). The two trends may imply that the smaller droplets below the critical pressure would take longer to evaporate than the larger droplets above the critical pressure. This apparent contradiction is resolved by considering the effect of clusters on the mass transport at the droplet surface.

It is impractical to consider all the interactions among the atoms in a submicron-size droplet in a dense background fluid to evaluate the evaporation timescale. Instead, a submicron-size droplet is large enough to regard the evaporation as a surface process. Thus, we adopt an evaporation model similar to that of subcritical fluids to explain our experimental results[34,35].

The prolonged survival over an hour implies that the droplet and the surrounding fluid are in a quasi-thermal equilibrium and that the mass influx $\Gamma_\mathrm{in}$ would balance the mass outflux $\Gamma_\mathrm{out}$ in the transition layer around the droplet surface (Fig. 6), indicating that the net mass flux $\Gamma_\mathrm{net}$ is negligible:

$$\begin{aligned} \Gamma_\mathrm{net} &= \Gamma_\mathrm{out} - \Gamma_\mathrm{in} \\ &= v_\mathrm{out}\rho_\mathrm{out} - v_\mathrm{in}\rho_\mathrm{in} \approx 0 \end{aligned} \qquad (7)$$

where $v_\mathrm{in(out)}$ and $\rho_\mathrm{in(out)}$ are the surface-normal mean velocities and mass densities, respectively. The outflux is defined as the flux leaving the inner boundary of the transition layer, i.e.,

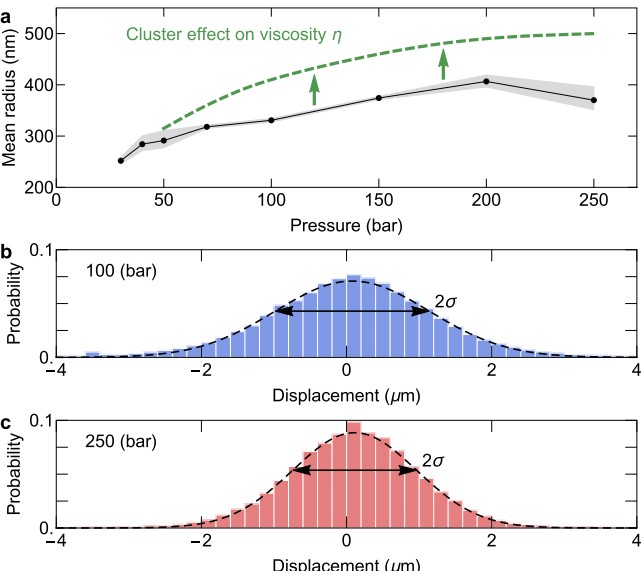

**Fig. 5 Mean size of droplets at different pressures. a** Brownian motion analysis is adopted to measure the mean size at each pressure. The droplet motions are split into two axes, which gives two sets of one-dimensional displacements. The medium viscosity tends to decrease with the appearance of the clusters. The green dashed line depicts the change in estimated droplet sizes, taking into account the cluster effect on the viscosity in a situation where the cluster density is high as in our experiments. The error band shows 90% confidence interval of normal distribution fit for each pressure. **b**, **c**. The histograms show the displacement of droplets during the elapsed time (0.02 s) and their normal distribution fits at 100 bar and 250 bar, respectively.

the surface of the droplet. The surface-normal mean velocity is then

$$v_\mathrm{in\,(out)} = \sqrt{\frac{8k_\mathrm{B} T}{\pi m_\mathrm{in\,(out)}}} \qquad (8)$$

where $m_\mathrm{in\,(out)}$ denotes the mass of the mass carrier. Assuming that the droplet has the density of liquid argon ($\rho_\mathrm{out} = \rho_\mathrm{droplet} = \rho_\mathrm{liquid}$), the net mass flux becomes

$$\Gamma_\mathrm{net} \approx \sqrt{\frac{8k_\mathrm{B} T}{\pi}} \left( \frac{\rho_\mathrm{droplet}}{\sqrt{m_\mathrm{out}}} - \frac{\rho_\mathrm{medium}}{\sqrt{m_\mathrm{in}}} \right) \qquad (9)$$

where the mass density of influx $\rho_\mathrm{in} = \rho_\mathrm{medium}$. When the net flux stays negligible, the evaporation timescale would be extended.

During compression, the number of clusters in the medium rapidly increases when crossing the critical pressure, and the average $m_\mathrm{in}$ increases. Thereby, the influx significantly reduced, and as a result, the evaporation timescale shortened (the region around the critical pressure in Fig. 3b). On the other hand, under subcritical pressures, the number of clusters is fewer; therefore, the influx is significant, resulting in a much longer evaporation timescale (the higher pressure region in Fig. 3b). If the pressure continues to increase and exceeds the critical pressure, the number of clusters also increases. However, as $\rho_\mathrm{medium}$ gradually approaches $\rho_\mathrm{droplet}$, the overall flux is rebalanced, resulting in a prolonged evaporation timescale.

## Discussion

The size estimation by the Brownian motion analysis obtained through Eq. (6) depends on the accuracy of the medium viscosity. In our study, we used the viscosity value corresponding to a

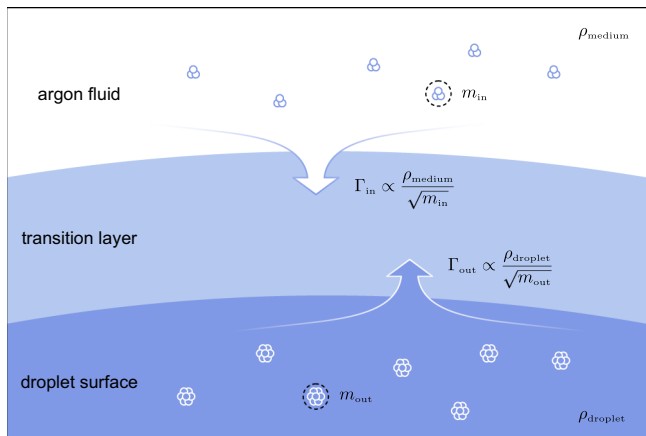

**Fig. 6 Graphical illustration of the mass transport at a droplet surface.** The emergence of clusters reduces the mass influx, and consequently, the evaporation time of a droplet decreases.

homogeneous argon fluid at the given pressure and temperature, taken from the NIST Chemistry WebBook[24]. However, as the pressure increases, more clusters are generated in the medium, and it is necessary to understand how this will affect the viscosity of the fluid. Instead of a comprehensive analysis of the viscosity, a simplified analysis is sufficient to show that the viscosity of the medium containing clusters would be lower than that of the homogeneous medium.

According to the kinetic model, the viscosity $\eta$ of a medium consisting of spherical particles of radius r and mass $m$ is

$$\eta \propto \frac{\sqrt{mT}}{\pi r^2} \qquad (10)$$

where $T$ is the temperature of the medium[36]. When considering the interaction between particles, the proportionality constant may have a temperature dependence, but the fundamental relationship does not change significantly. Assuming that each cluster consists of $N$ argon atoms on average and the temperature is fixed,

$$\eta \propto N^{-1/6} \qquad (11)$$

as $m \propto N$ and $r \propto N^{1/3}$. Thus, the viscosity of the medium decreases if atoms aggregate to form clusters; consequently, the droplet size is underestimated for a higher cluster density. This cluster effect on the estimation of droplet size is illustrated by the green dashed curve in Fig. 5.

In summary, we demonstrate that the prolonged survival of droplets in SCF is possible. The analysis of mass transport shows that the clustering can reduce the lifetime of droplets, indicating that the clustering effect plays an essential role in the understanding of nonequilibrium thermodynamic processes in supercritical states.

The existence of quasi-steady clusters and droplets will have practical implications in various fields of science and engineering involving SCFs such as planetary meteorology, power plant cooling systems, pharmaceutical processes, high-power switching, and high-pressure fuel injection. For instance, the clusters and droplets may affect the mass transport processes in the dense atmospheres of Venus and Jupiter. The same effect may have to be considered in the recent development of the SCF $CO_2$ cleaning technique in semiconductor fabrication. We expect our findings to serve as an important milestone in exploring the nonequilibrium, transient behavior of SCFs. The physics underlying clusters and droplets in SCFs will be an immediate topic of interest in a wide range of engineering problems related to heat and momentum transport processes in SCFs.

## Methods

**High-pressure chamber system.** The reciprocating compressor has a cycling frequency of about 4.1 Hz and a throughput of about $6.8\,cm^3\,cycle^{-1}$ (under standard conditions). A pair of bendable stainless-steel hoses (Swagelok SS-FX4TA4TA4) is installed to add flexibility to the system. Quick-connect adapters (SS-QF4-S, SS-QF4-B) are placed at the junctions, which modularize the system and thereby make it possible to carry the chamber after compression. The cubic-shaped stainless-steel chamber (side length: 65 mm) is computer numerical control machined to make six threading ports (1 1/16″-12-UN) with O-ring grooves (SAE J1926-1). One of the ports is used as the gas entrance, and sight windows are installed on the other ports. The sight windows, made of sapphire glass (thickness: 7.2 mm) and bonded in the stainless-steel housing, has a clear aperture 11.2 mm in diameter (Rayotek 101117C). Two different pressure sensors, one with a digital display (Keller LEO2) and the other with an analog voltage output (Tival Sensors TST-20) are installed to monitor the pressure for safety. The high-pressure chamber can preserve the internal pressure securely for a long time (see Supplementary Fig. 1). A thermocouple (Fluke 80PK-27) is used to measure the temperature of the expanding fluids.

**Laser scattering.** A vertically polarized continuous He–Ne laser at 633 nm with 5 mW output power (Thorlabs HNL050LB) operates as the scattering source. Two lenses manipulate the beam profile of the incident laser: a plano-convex lens with a focal length of 50 mm (Thorlabs LA4148) and a laser line generator (or Powell lens) with a fan angle of 23° (Thorlabs FLG10FC-633). Right-angle scattered signals are captured by ICCD (Princeton Instruments PI-MAX4) with imaging optics. The imaging optics system enables ×2.5 and ×11 magnification through a single lens system with different optical lengths embedded into the lens tube to avoid ambient light (Thorlabs LA4148).

**Particle tracking code.** Mathematica (Wolfram Research 12.0 Student Edition) is used to analyze the images and videos. Two built-in functions (ComponentMeasurements and ImageFeatureTrack) are utilized to determine the number density of droplets from the images and the Brownian motion of droplets from the videos.

**COMSOL Multiphysics simulation.** We calculate the expansion of an ideal fluid through the high Mach number flow physics in COMSOL Multiphysics®. The software version is 5.5 with a CFD module. It calculates the momentum and mass continuity equations and the temperature equation. The temperature equation includes the contributions of viscous heating and pressure work. The effects related to the phase change are not considered since an ideal gas is assumed. Sutherland's law gives the temperature dependence of thermal conductivity and kinematic viscosity. 2D axisymmetric geometry is adopted to reduce the computational load. Also, rounding sharp edges helps to avoid discontinuous change in the calculation. The mesh size is larger than 0.5 mm to avoid the CFD error while treating the high fluid speed during the expansion. The mesh size limits the space resolution but still gives an adequate temperature trend.

## Data availability

All relevant data supporting this study are available from the authors (S.L. and G.Y.) upon reasonable request.

## Code availability

All codes are available from the authors (S.L. and G.Y.) upon reasonable request.

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

## Acknowledgements

This study is supported by the National Research Foundation of Korea (NRF) funded by the Ministry of Science and ICT (No. 2019R1A2C3011474 and 2016K1A4A4A01922028) and Ministry of Education (No. 2019R1A6A3A13091407).

## Author contributions

G.Y. proposed the original idea and conceived the project. S.L. and J.L. designed the high-pressure chamber system and performed the experiments. S.L. carried out the majority of the data processing and wrote the draft of the manuscript. S.L. and G.Y. analyzed the data of the Brownian motions and developed the pseudo-evaporation model. Y.K. and S.J. performed the molecular dynamics simulations to assist in the interpretation of the pseudo-evaporation model. S.J. confirmed the temperature drop during the fluid expansion by COMSOL Multiphysics simulation. G.Y. and D.K. supervised the project. All authors contributed significantly to the discussion of the results and the writing of the manuscript.

## Competing interests

The authors declare no competing interests.
