## [Peer Review File · Nature Communications]

REVIEWER COMMENTS

Reviewer #1 (Remarks to the Author):

The paper "Quasi-equilibrium phase separation in supercritical fluids" reports experiments on supercritical Ar at the temperature about twice the critical one and in a pressure range between 30 and 300 bar, i.e. on the low-pressure side from the Widom line. The main finding is in observation of the decaying Rayleigh-Mie scattering signals in the supercritical Ar, which the authors ascribe to evaporating liquid droplets created during the filling the chamber with Ar. The authors estimated the medium opacity of the system and claimed it is mainly due to some small clusters produced by compression-expansion processes.

Although the experimental observations look interesting, I was not convinced that this is the signal from the non-equilibrium droplets. Moreover, the authors were not able to observe the droplets by direct imaging. It is not clear why "the fluid undergoes a sudden expansion and cools to form liquid-like droplets and clusters in the process of filling the chamber with argon" as proposed by the authors. Do they mean the local temperature can drop below the critical one? Is it possible to simulate this process by computer simulations? I do not see here any solid justification for the suggested mechanism of droplet formation.

Another unclear issue is the possibility of existence of small clusters. Why in the supercritical low-density state there should exist clusters of Ar atoms is not explained - only it is stated that they are not spontaneous but created "by a compression-expansion process". No clusters were found in supercritical Ar in numerous equilibrium computer simulations - however the authors believe that the non-equilibrium conditions can form them, that is just a suggestion without any solid proof.

Other minor comments:

- Eq.2 is the Einstein mean square displacement for one-dimensional motion $\langle dx^2 \rangle$.
- perhaps the authors did not read carefully Ref.[12], which contains mistakes when for the energy integral with Debye density of vibrational states (coming from dispersion relation of acoustic excitations) one applies the model of harmonic oscillators with the same energy, that is nonsense.

Reviewer #2 (Remarks to the Author):

Manuscript: Nature Communications manuscript NCOMMS-20-25905

Title: Quasi-equilibrium phase separation in supercritical fluids

Author(s): Seungtaek Lee, Juho Lee, Yeonguk Kim, Seokyong Jeong, Dong Eon Kim, Gunsu Yun

The authors present observations of phase separation in argon in the state of supercritical fluid (SCF), which the authors claim as a "pseudo phase transition" between liquid-like (LL) and gas-like (GL) phases similarly to the gas-liquid phase transition across the coexistence line in subcritical fluids. The authors stress an analogy between the phase transition in subcritical and supercritical regimes of the phase diagram: "LL-GL phase separation is possible by generating submicron size LL argon droplets in a GL argon SCF", and characterize the GL fluid as a "quasi-equilibrium clustered state well above the critical temperature, with a significant increase in cluster formation rate traversing the critical pressure".

Comments:

Line 32-41: The authors claim that no phase separation occurs in SCFs under equilibrium conditions and define the Widom line as a "pseudo" phase transition line between gas-like (GL) and liquid-like (LL) regimes.

This is not correct as the Widom line represents extrema of thermodynamic response functions, whereas the Frenkel line defines the borderline between GL and LL at slightly larger pressure fields. Originally, the transition of GL to LL phase was defined on basis of changes of the dynamic process of diffusion as discussed in many papers such as by Bolmatov et al in Ref [12]. Another paper by Pipich et al. cited in Ref. [14] observes droplet formation above the Frenkel line in SC-CO₂ and shows a clear distinction between Widom and Frenkel line. Quite recently, the same authors published an extension of this work in Scientific Reports. Both papers using small-angle neutron technique show droplet formation above the Frenkel as well as the gas-liquid phase boundary under equilibrium condition.

General questions and comments: The authors observe droplet formation and evaporation of 0.3 to 0.4 μm radius and smaller droplets (they call them cluster) in the gas-like regime below the Widom line. These observations are somewhat mysterious to me and raises several questions:

1. The droplets form during the filling procedure of the cell at RT with argon up to a pressure of 250 bar taking about 10 minutes (Fig. 2a). The authors controlled the pressure with two pressure sensors but seemingly not the temperature, as they did not say anything about a decline of temperature. Could the temperature become small enough to reach the liquid regime? I cannot imagine but the authors have to discuss this issue.
2. Fig. 1d shows the pathway of the experiment near $T/T_C = 2$. At this temperature the authors claim ideal gas condition due to their estimation of $Z < 1$. However, an ideal gas cannot show supercritical behavior and phase separation!
3. An interesting observation is the strong enhancement of droplet number density and cluster formation (opacity) in Fig. 2 a) and c)) above the critical pressure. There cannot be any influence of critical pressure at a temperature twice as large as the critical one! This observation, however, might give a hint about the degree of temperature decline and influence of the critical point during filling.
4. The authors claim that thermal density fluctuations do not occur above twice the critical temperature. What is the basis of this statement? Because of ideal gas conditions?
5. The authors name the droplet formation of larger and smaller droplets as "pseudo-phase" separation and "quasi-steady clustering", respectively, occurring in the gas-like regime of argon SCF. These names appear to me as undefined empty phrases.
6. As already mentioned, the formation of droplets during the filling process of the chamber appears mysterious. One needs the pressure-temperature pathway during the filling process. The low surface energy in SCFs could explain the stability of the LL droplets later under conditions given by the red points in Fig.1d. From this viewpoint, the droplets are in metastable condition well defined in the theory of phase transition. On this basis, I see no need for phrases of "pseudo-phase separation" and "quasi-steady clustering".

General summary:

The authors have to substantially clarify their manuscript and clearly describe the experimental conditions of the filling procedure. I recommend the editor not to accept this manuscript.

Reviewer #3 (Remarks to the Author):

The study describes a phase separation phenomenon in supercritical fluids and demonstration that LL-GL phase separation is possible by generating submicron size LL argon droplets in a GL argon SCF. The manuscript is interesting but because of the not very high quality I do not recommend it for publication in this very high quality journal.

RESPONSE for the REVIEWER COMMENTS

>> We thank the reviewers for their detailed comments and constructive criticisms on our work. We have conducted new experiments, analyzed the results with help of numerical simulations, and expanded the survey of relevant literature to address the comments reviewers. The following are our point-by-point responses to the comments and criticisms of the referees. The corresponding changes are highlighted in the revised manuscript.

Reviewer #1 (Remarks to the Author):

The paper "Quasi-equilibrium phase separation in supercritical fluids" reports experiments on supercritical Ar at the temperature about twice the critical one and in a pressure range between 30 and 300 bar, i.e. on the low-pressure side from the Widom line. The main finding is in observation of the decaying Rayleigh-Mie scattering signals in the supercritical Ar, which the authors ascribe to evaporating liquid droplets created during the filling the chamber with Ar. The authors estimated the medium opacity of the system and claimed it is mainly due to some small clusters produced by compression-expansion processes.

>> We appreciate the correct summary of our manuscript followed by detailed comments and constructive criticisms.

(1) Although the experimental observations look interesting, I was not convinced that this is the signal from the non-equilibrium droplets. Moreover, the authors were not able to observe the droplets by direct imaging. It is not clear why "the fluid undergoes a sudden expansion and cools to form liquid-like droplets and clusters in the process of filling the chamber with argon" as proposed by the authors. Do they mean the local temperature can drop below the critical one? Is it possible to simulate this process by computer simulations? I do not see here any solid justification for the suggested mechanism of droplet formation.

>> [Page 3-5, Fig.2] We added one subsection (Result – Temperature drop of expanding fluids) to justify the formation of droplets and clusters. We show that the expanding fluid along a tube through an orifice from a pressurized state can reach a low enough temperature to form liquid droplets. The local temperature drop is confirmed in 2D ideal gas simulation and compared with the experiment.

(2) Another unclear issue is the possibility of existence of small clusters. Why in the supercritical low-density state there should exist clusters of Ar atoms is not explained - only it is stated that they are not spontaneous but created "by a compression-expansion process". No clusters were found in supercritical Ar in numerous equilibrium computer simulations - however the authors believe that the non-equilibrium conditions can form them, that is just a suggestion without any solid proof.

>> [Page 6, Eq.1-2, Fig.4] We thank the reviewer for pointing out that the existence of small clusters has not been observed in equilibrium computer simulations. We take this question as an opportunity to better illustrate that our observations capture an unexpected non-equilibrium aspect of supercritical fluids, namely droplets and small clusters of long time scale. The existence of small clusters is explained further at the end of the subsection "Result – Formation of droplets and clusters": We set a lower bound on the cluster number density for a range of cluster size based on the opacity measurement of our supercritical fluids. The droplets and clusters are generated by local temperature drop during the compression, and they evaporate with a long time scale (~ 1 hour). To clearly deliver the fact that the clusters are NOT generated spontaneously but rather formed by a compression-expansion process, we have added an explanation on the local temperature drop before the description on the clusters' size and density estimation.

Other minor comments:

- Eq.2 is the Einstein mean square displacement for one-dimensional motion $\langle dx^2 \rangle$.

>> [Page 7, Eq.3, Fig.5] During the data processing, the droplet motions (which are projected to 2D) are split into two axes (vertical and horizontal), which gives two sets of one-dimensional displacements, and Eq.2 is applied. We prefer this for easy correction of the origin. To eliminate the ambiguity, we added a short description to the caption of Fig.5.

- perhaps the authors did not read carefully Ref.[12], which contains mistakes when for the energy integral with Debye density of vibrational states (coming from dispersion relation of acoustic excitations) one applies the model of harmonic oscillators with the same energy, that is nonsense.

>> We appreciate your comment for better understanding of the literature. Remove it?

Reviewer #2 (Remarks to the Author):

The authors present observations of phase separation in argon in the state of supercritical fluid (SCF), which the authors claim as a “pseudo phase transition” between liquid-like (LL) and gas-like (GL) phases similarly to the gas-liquid phase transition across the coexistence line in subcritical fluids. The authors stress an analogy between the phase transition in subcritical and supercritical regimes of the phase diagram: “LL-GL phase separation is possible by generating submicron size LL argon droplets in a GL argon SCF”, and characterize the GL fluid as a “quasi-equilibrium clustered state well above the critical temperature, with a significant increase in cluster formation rate traversing the critical pressure”.

>> We appreciate the succinct summary of our manuscript followed by constructive criticisms.

Comments:

Line 32-41: The authors claim that no phase separation occurs in SCFs under equilibrium conditions and define the Widom line as a “pseudo” phase transition line between gas-like (GL) and liquid-like (LL) regimes. This is not correct as the Widom line represents extrema of thermodynamic response functions, whereas the Frenkel line defines the borderline between GL and LL at slightly larger pressure fields. Originally, the transition of GL to LL phase was defined on basis of changes of the dynamic process of diffusion as discussed in many papers such as by Bolmatov et al in Ref [12]. Another paper by Pipich et al. cited in Ref. [14] observes droplet formation above the Frenkel line in SC-CO₂ and shows a clear distinction between Widom and Frenkel line. Quite recently, the same authors published an extension of this work in Scientific Reports. Both papers using small-angle neutron technique show droplet formation above the Frenkel as well as the gas-liquid phase boundary under equilibrium condition.

>> [Page 2, Introduction] The definitions and the recent studies related to the Frenkel line and the Widom line are included in the “Introduction” section of the revised manuscript. As the referee correctly states, the Widom line is the locus of the extrema of thermodynamic response functions in the supercritical regime. The Widom line is often described as an extension of the coexistence line (where the thermodynamic

response functions “diverge”) in the subcritical regime. In this sense, we used the phrase “pseudo” phase transition between GL and LL states.

On the other hand, the Frenkel line originates from the particle picture (as opposed to the continuum picture) and separates the supercritical fluids (in equilibrium) into rigid state and non-rigid state. The changes of physical properties across the Frenkel line seem more conspicuous than those across the Widom line, and we note that the terms “non-rigid and rigid” are generally ascribed to the states separated by the Frenkel line rather than the terms “GL and LL”. However, we also recognize that the term ‘liquid-liquid’ phase transition is also adopted to describe the droplet formation well above the Widom line in SC-CO₂ reported by Pipich and Schwahn [Sci. Reports 2020; Ref. 18 in the revised manuscript]. We believe that this choice was to make a connection with the concept of polymorphic phase transition of one-component disordered systems such as water and glasses.

To avoid confusion, we revised the manuscript to emphasize that the droplets and clusters in our work corresponds to novel non-equilibrium features that are produced by the dynamic compression-expansion process and maintained for a long time scale (~1 hour) whereas the notions of the Widom line and the Frenkel line are established for the equilibrium conditions. The distinction of GL and LL in our work refers to the difference in mass density that were manifested in the optical scattering measurements [cf. Refs. 22 and 23 in the revised manuscript].

General questions and comments: The authors observe droplet formation and evaporation of 0.3 to 0.4 μm radius and smaller droplets (they call them cluster) in the gas-like regime below the Widom line. These observations are somewhat mysterious to me and raises several questions:

1. The droplets form during the filling procedure of the cell at RT with argon up to a pressure of 250 bar taking about 10 minutes (Fig. 2a). The authors controlled the pressure with two pressure sensors but seemingly not the temperature, as they did not say anything about a decline of temperature. Could the temperature become small enough to reach the liquid regime? I cannot imagine but the authors have to discuss this issue.

>> [Page 3-5, Fig.2] We have conducted experiments and simulations to measure the local temperature of the argon fluid expanding through the tube and confirmed that the

local temperature drops below the critical temperature. Please see our response to the question 1 of the reviewer #1.

2. Fig. 1d shows the pathway of the experiment near $T/T_C = 2$. At this temperature the authors claim ideal gas condition due to their estimation of $Z < 1$. However, an ideal gas cannot show supercritical behavior and phase separation!

>> [Page 3, Fig.1d] We eliminated the label “ideal gas” in Fig.1d, which is misleading. Under the homogeneous and equilibrium condition, the argon fluid holds the compressibility factor $Z \sim 1$, along the experimental conditions, which becomes exactly unity for an ideal gas. However, in our experiment, the argon fluid undergoes a non-equilibrium process resulting in the inhomogeneous state characterized by clusters and droplets and takes a long time to reach an ultimate equilibrium state. In this sense, Fig.1d serves as an approximate reference for the states involved in our experiments. To make this point clear, we corrected the caption of Fig.1d to “Extended phase diagram of homogeneous and equilibrium argon fluid”.

3. An interesting observation is the strong enhancement of droplet number density and cluster formation (opacity) in Fig. 2 a) and c)) above the critical pressure. There cannot be any influence of critical pressure at a temperature twice as large as the critical one! This observation, however, might give a hint about the degree of temperature decline and influence of the critical point during filling.

>> [Page 5, Fig.3] Our present manuscript focuses on the existence of the droplets and clusters in the supercritical argon fluid sufficiently away from the critical point and on the evaporation process of the droplets. It is indeed a mystery why the droplet and cluster generation rates drastically increase when crossing the critical pressure, which will be the topic of our future research.

4. The authors claim that thermal density fluctuations do not occur above twice the critical temperature. What is the basis of this statement? Because of ideal gas conditions?

>> [Page 3, Result – Experimental apparatus and conditions (1st paragraph)] The critical phenomena such as a density fluctuation in the supercritical fluids are dramatic near the critical point [Ref. 3 in the manuscript]. It is known that a spontaneous and

persistent density fluctuation to form the clusters does not occur when the fluid is far from the critical point.

Note that the theory of critical phenomena assumes the fluid in thermal equilibrium. In contrast, in our experiment, the clusters are generated during the compression-expansion process. The corresponding statement in the first paragraph of the section “Result” is revised to emphasize this difference; “Because the temperature is about two times the critical temperature, spontaneous critical phenomena such as density fluctuations do not occur; however, the fluid undergoes a sudden expansion and adiabatically cools to form a liquid during the compression.”

5. The authors name the droplet formation of larger and smaller droplets as “pseudo-phase” separation and “quasi-steady clustering”, respectively, occurring in the gas-like regime of argon SCF. These names appear to me as undefined empty phrases.

>> [Page 1, Introduction] The words “pseudo-phase” and “quasi-steady” are used to distinguish our observations from the phase transitions in the subcritical regime and the ideal equilibrium states. As the droplets and clusters persist for an hour and eventually evaporate (they survive for a long time but not eternally), it is different from the equilibrium phase separation. However, we admit that the words like pseudo and quasi are often overused in scientific literature. We examined that our case might fall into such category by surveying more literature. With some reservations, we refined the terminologies as follows:

Before	Revised
LL droplets	(liquid) droplets
pseudo-evaporation	evaporation
pseudo phase transition	
quasi-steady	quasi-steady (preserved)

We noticed that in the field of high-pressure flow engineering [cf. Ref. XX: J. Bellan] the term ‘emission’ is preferred instead of the term ‘evaporation’ because the latter technically means ‘turning from liquid to vapor’ and may give misleading information on the phase changes in the supercritical fluids. However, we believe that the meaning of evaporation can be broadened in our context for the sake of frugality of words. We

devoted a paragraph to define and explain the terms above in the introduction section of the revised manuscript.

Note also that we have modified slightly the title to “Quasi-equilibrium phase separation in single-component supercritical fluids”. This is to avoid possible confusion with the phase separation phenomena in mixtures.

6. As already mentioned, the formation of droplets during the filling process of the chamber appears mysterious. One needs the pressure-temperature pathway during the filling process. The low surface energy in SCFs could explain the stability of the LL droplets later under conditions given by the red points in Fig.1d. From this viewpoint, the droplets are in metastable condition well defined in the theory of phase transition. On this basis, I see no need for phrases of “pseudo-phase separation” and “quasi-steady clustering”.

>> [Page 3, Fig.1d] The pressure-temperature pathway during the compression-expansion process is indicated in the revised Fig.1d. The fluid undergoes an adiabatic expansion for the earlier compression stage (when the chamber pressure is below ~ 100 bar) and locally reaches a low enough temperature to form a liquid.

In the kinetic point of view, we have suggested that the prolonged evaporation time of a droplet is explained by considering the clustering effect on the mass transport at the droplet surface.

As the reviewer suggests, the stability of the droplets may also be analyzed in the view of the continuum thermodynamics. Adopting the usual Young-Laplace model that describes the nucleation threshold in the subcritical fluids, we may define the change of the grand potential for a system with a spherical GL droplet of radius r in a LL background (assuming fixed chemical potential, temperature, and volume) as follows [W. Sung, “Statistical Physics for Biological Matter”, Graduate Texts in Physics series, 2018 Springer]:

$$\Delta\Omega = -\frac{4\pi r^3}{3}\Delta p + 4\pi r^2\gamma$$

where γ is the surface tension and $\Delta p = p_{LL} - p_{GL}$ is the pressure difference between the nucleus (GL) and the background (LL). Here it is assumed that p_{LL} does not depend on the nucleus size. The threshold of this potential for the formation of an GL droplet is

$$\Delta\Omega_c = \frac{16\pi \gamma^3}{3 \Delta p^2}$$

This equation may imply that the low surface tension in SCFs would make it easier for a nucleus to transform to a LL droplet. However, the growth of the nucleus corresponds to a permanent phase change. In order to have a metastable state, Δp would need to have a dependence on the nucleus size. For example, $\Delta p(r) \sim \frac{1}{1+\exp(r-r_0)}$ would give a shallow local minimum in the potential as illustrated in the graph below. We decided not to include this thermodynamic argument in the manuscript because the size dependence of $\Delta p(r)$ is not known.

General summary:

The authors have to substantially clarify their manuscript and clearly describe the experimental conditions of the filling procedure. I recommend the editor not to accept this manuscript.

>> We have run additional experiments and simulations to clarify the thermodynamic pathway of the argon fluid. Those results together with the improved analysis of the

scattering data reinforce that the long-lived LL droplets and clusters are novel non-equilibrium properties of supercritical fluids. We have substantially revised our manuscript to emphasize these points.

Reviewer #3 (Remarks to the Author):

The study describes a phase separation phenomenon in supercritical fluids and demonstration that LL-GL phase separation is possible by generating submicron size LL argon droplets in a GL argon SCF. The manuscript is interesting but because of the not very high quality I do not recommend it for publication in this very high quality journal.

>> We have substantially revised our manuscript to better describe the novel non-equilibrium phenomena (the long-lived LL droplets and clusters) in the supercritical fluids and put them in a clearer context by comparing with the recent development of equilibrium phase separation phenomena (i.e., the Frenkel and Widom lines).

REVIEWER COMMENTS

Reviewer #1 (Remarks to the Author):

I have read the revised version of "Quasi-equilibrium phase separation in supercritical fluids" and the authors' rebuttal to the comments of all referees. In my opinion the authors reasonably replied to the critical comments and made several corresponding changes.

However, I have to stress once again - making references to papers with wrong claims/results is equivalent to their dissemination. Perhaps, these authors do not realize what is behind the title of Ref.[14] ("Liquid-Gas" Transition in the Supercritical Region ...). Do the authors agree, that the standard gas-liquid separation below the critical point T_c (binodal) drastically shifts with discontinuity to the higher densities above T_c in order to coincide with the suggested in Ref.14 "Liquid-Gas Transition in the Supercritical Region"??? Making reference to [14] is the same as to agree to this nonsense. The same can be said about Ref.[16] which I mentioned in my previous report. All this "Frenkel line" concept does not have any rigid theoretical basis and is rather a speculation. I would recommend the authors to verify any results on the "Frenkel line".

A minor revision is required.

Reviewer #2 (Remarks to the Author):

2nd Answer to Manuscript Number: advs.202002312

Title: Quasi-equilibrium phase separation in single-component supercritical fluids

Authors: Florentina Maxim et al.

The authors satisfactory answered the comments and questions of this Reviewer and substantially clarified their manuscript in particular emphasizing that the droplets and clusters in their work corresponds to a non-equilibrium situation produced by a compression-expansion process in contrast to equilibrium conditions so far studied near the Widom and Frenkel line. From this perspective this work appears to me novel and relevant what makes me to suggest the editor to publish this work as it.

RESPONSE for the REVIEWER COMMENTS

>> We thank the reviewers for their affirmative comments and constructive criticisms of our work. We have expanded the survey of relevant literature to address the comments of Reviewer #1. The following are our point-by-point responses.

Reviewer #1 (Remarks to the Author):

I have read the revised version of "Quasi-equilibrium phase separation in supercritical fluids" and the authors' rebuttal to the comments of all referees. In my opinion the authors reasonably replied to the critical comments and made several corresponding changes.

>> We appreciate the positive comments on the revision, where we have taken the Reviewer's constructive criticisms as an opportunity to clarify our findings as novel non-equilibrium aspects of supercritical fluids (SCFs).

However, I have to stress once again - making references to papers with wrong claims/results is equivalent to their dissemination. Perhaps, these authors do not realize what is behind the title of Ref.[14] ("Liquid-Gas" Transition in the Supercritical Region ...). Do the authors agree, that the standard gas-liquid separation below the critical point T_c (binodal) drastically shifts with discontinuity to the higher densities above T_c in order to coincide with the suggested in Ref.14 "Liquid-Gas Transition in the Supercritical Region"??? Making reference to [14] is the same as to agree to this nonsense. The same can be said about Ref.[16] which I mentioned in my previous report. All this "Frenkel line" concept does not have any rigid theoretical basis and is rather a speculation. I would recommend the authors to verify any results on the "Frenkel line". A minor revision is required.

>> Regarding ref. [16] (which is eliminated in the revised manuscript):

We now recognize the mistake in the derivation of eqn. (2) from eqn. (1) in ref. [16] (D. Bolmatov et al., Nat. Commun., 2013). Although the authors stated that the approximation only holds when $\hbar\omega \ll k_B T$, the phonon energy scale ($\hbar\omega$) appears not to satisfy this criterion according to figure 2 in ref. [11] (G. Simeoni et al., Nat. Phys., 2010). Thus, we have removed ref. [16] in the revised manuscript.

>> Regarding ref. [14] (which has been changed to ref. [15] in the revised manuscript):

We did not intend to agree to all the contents in ref. [14] by citing the paper, but rather, we intended to introduce the concept of the Frenkel line and explain how it is used to describe the emergent states in equilibrium SCFs. After re-reading the cited papers and expanding our literature survey, we found healthy the controversy and debates about the relevance of the Widom line and the Frenkel line to the phase transitions in equilibrium SCFs. Thus, it will be in our best interest to keep the healthy

debates on neutral ground until future studies resolve the controversy. In this spirit, we have relocated the citations and brought in two more references as ref. [17, 18] in the 3rd paragraph in the introduction section of the revised manuscript. A series of refs. [15–18] covers the controversy by two leading authors, T. Bryk and V. Brazhkin.

Note that our paper mainly concerns the non-equilibrium aspects of SCFs, which appear to be remotely related to the controversy. Nonetheless, we hope that our work also provides insights into the emergent phases in equilibrium SCFs.

Reviewer #2 (Remarks to the Author):

The authors satisfactorily answered the comments and questions of this Reviewer and substantially clarified their manuscript in particular emphasizing that the droplets and clusters in their work corresponds to a non-equilibrium situation produced by a compression-expansion process in contrast to equilibrium conditions so far studied near the Widom and Frenkel line. From this perspective this work appears to me novel and relevant what makes me to suggest the editor to publish this work as it.

>> We appreciate the Reviewer for recognizing the central aspect of our findings and providing constructive comments during the review process.

REVIEWER COMMENTS

Reviewer #1 (Remarks to the Author):

I do not have some essential comments to the last revised version of "Quasi-equilibrium phase separation in supercritical fluids". However, my feeling is that the authors do not really understand the effects of atomic motion in fluids on the shape of time correlation functions. And therefore rebroadcast from the literature not really correct definitions. For example, in lines 42-43 they write "... locus in the phase diagram where the velocity autocorrelation function signifies the presence of an oscillatory component [15]". What do the authors mean under the "oscillatory component"? How do they discriminate between the particle oscillatory motion and a single back-scattering by a neighbor atom? The latter occurs even for very low-density fluids. And it has nothing in common with oscillatory motion around some energy minimum.

Furthermore, the title of [15] in line 374-375 sounds "Liquid-Gas" Transition in the Supercritical Region: Fundamental Changes in 374 the Particle Dynamics" and the readers would believe, that the claimed emergence of the "presence of an oscillatory component" in the velocity autocorrelation function corresponds to the "Liquid-Gas" Transition in the Supercritical Region", that is absolutely wrong. It seems the authors believe too in the strange "Liquid-Gas Transition" in supercritical region which coincides with the "Frenkel line". I would recommend the authors to draw for themselves a density-temperature phase diagram with the standard binodal ending at the critical point with T_c , and add to that picture schematically the "Frenkel line". If the authors had any idea how the standard gas-liquid separation curve changed to the claimed in [15] "Liquid-Gas" Transition in the Supercritical Region - please explain. Because my point is - this does not make sense.

Another revision is required.

RESPONSE for the REVIEWER COMMENTS

>> We thank the reviewer for the affirmative comments and constructive criticisms of our work. We have expanded the survey of relevant literature to address the comments of Reviewer #1. The following are our point-by-point responses.

Reviewer #1 (Remarks to the Author):

I do not have some essential comments to the last revised version of "Quasi-equilibrium phase separation in supercritical fluids". However, my feeling is that the authors do not really understand the effects of atomic motion in fluids on the shape of time correlation functions. And therefore rebroadcast from the literature not really correct definitions. For example, in lines 42-43 they write "... locus in the phase diagram where the velocity autocorrelation function signifies the presence of an oscillatory component [15]". What do the authors mean under the "oscillatory component"? How do they discriminate between the particle oscillatory motion and a single back-scattering by a neighbor atom? The latter occurs even for very low-density fluids. And it has nothing in common with oscillatory motion around some energy minimum.

>> We have taken the Reviewer's constructive criticisms as an opportunity to correct our understanding. As the Reviewer pointed out, the change in the feature of the autocorrelation function cannot be used as an indicator of a phase change in SCF. It is already reported by Fisher & Widom in 1969: "It should perhaps be emphasized here that the abrupt change in the asymptotic character of $G(r)$ that occurs when the transition locus is crossed does not imply any thermodynamic singularities." (*J. Chem. Phys.* **50**, 3756 (1969))

Furthermore, the title of [15] in line 374-375 sounds "Liquid-Gas" Transition in the Supercritical Region: Fundamental Changes in 374 the Particle Dynamics" and the readers would believe, that the claimed emergence of the "presence of an oscillatory component" in the velocity autocorrelation function corresponds to the "Liquid-Gas" Transition in the Supercritical Region", that is absolutely wrong. It seems the authors believe too in the strange "Liquid-Gas Transition" in supercritical region which coincides with the "Frenkel line". I would recommend the authors to draw for themselves a density-temperature phase diagram with the standard binodal ending at the critical point with T_c , and add to that picture schematically the "Frenkel line". If the authors had any idea how the standard gas-liquid separation curve changed to the claimed in [15] "Liquid-Gas" Transition in the Supercritical Region - please explain. Because my point is - this does not make sense. Another revision is required.

>> We have excluded the reference entitled "Liquid-Gas Transition in the Supercritical Region: Fundamental Changes in the Particle Dynamics" in the revised manuscript. Additionally, not to mislead the potential readers, we have modified the introduction to emphasize that the focus of our manuscript is the non-equilibrium dynamics in SCFs regarding LL drops in GL background. Thus

the revised manuscript does not scrutinize any concepts related to the separatrices of equilibrium SCFs.